# Maternal Nutrition and Neurodevelopment: A Scoping Review

**DOI:** 10.3390/nu13103530

**Published:** 2021-10-08

**Authors:** María Camila Cortés-Albornoz, Danna Paola García-Guáqueta, Alberto Velez-van-Meerbeke, Claudia Talero-Gutiérrez

**Affiliations:** Neuroscience Research Group (NEUROS), Centro Neurovitae, School of Medicine and Health Sciences, Universidad del Rosario, Bogotá 111221, Colombia; mariacam.cortes@urosario.edu.co (M.C.C.-A.); danna.garcia@urosario.edu.co (D.P.G.-G.); alberto.velez@urosario.edu.co (A.V.-v.-M.)

**Keywords:** maternal nutrition, neurodevelopment, cognitive abilities, neuropsychiatric disorders, nutrients

## Abstract

In this scoping review, we examined the association between maternal nutrition during pregnancy and neurodevelopment in offspring. We searched the Pubmed and ScienceDirect databases for articles published from 2000 to 2020 on inadequate intake of vitamins (B12, folate, vitamin D, vitamin A, vitamin E, vitamin K), micronutrients (cooper, iron, creatine, choline, zinc, iodine), macronutrients (fatty acids, proteins), high fat diets, ketogenic diets, hypercaloric diets, and maternal undernutrition. Some older relevant articles were included. The search produced a total of 3590 articles, and 84 studies were included in the qualitative synthesis. Data were extracted and analyzed using charts and the frequency of terms used. We concluded that inadequate nutrient intake during pregnancy was associated with brain defects (diminished cerebral volume, spina bifida, alteration of hypothalamic and hippocampal pathways), an increased risk of abnormal behavior, neuropsychiatric disorders (ASD, ADHD, schizophrenia, anxiety, depression), altered cognition, visual impairment, and motor deficits. Future studies should establish and quantify the benefits of maternal nutrition during pregnancy on neurodevelopment and recommend adequate supplementation.

## 1. Introduction

The role of maternal nutritional factors in fetal development emerged as an important research topic during the 20th century. Initial research on folate supplementation and its efficacy in reducing neural tube defects took place in the 1960s, setting the basis for recommendations regarding folate supplementation during pregnancy [1]. In the same decade, iron levels in pregnant women and the effects of oral iron supplementation were discussed, and eventually, researchers uncovered the role of iron deficiency anemia in fetal development [2]. Specifically, iron deficiency was associated with adverse outcomes, including preterm delivery, maternal mortality, and poor performance on developmental scales [2,3]. Afterward, relationships were defined between fetal development and additional nutritional elements, such as micronutrients and macronutrients, as the effects of deficiencies and supplementation of these elements were characterized with respect to long-term outcomes. This growing evidence for the importance of maternal nutrition was supported by Barker’s fetal programming hypothesis, which states that alterations in fetal nutrition have long-term effects on an individual’s health and predisposition to disease in adulthood [4].

Periconceptional and prenatal environments are critical for fetal brain development [4,5]. Maternal nutritional signals determine the epigenetic remodeling of fetal genes, and these effects persist through implantation, influencing placental development and nutrient transference [6]. Neurulation occurs early in embryonic life, beginning on day 18 after conception and continuing until day 28 [7]. A series of cellular events takes place, including neurogenesis from day 42 to mid-gestation, which is followed by migration, differentiation, synapse formation, and apoptosis [8]. Although brain development continues during the postnatal period, the fetal brain is especially susceptible to stimuli and insult due to rapid change, high sensitivity to the environment, and the prolonged duration of the brain developmental process [7].

Both experimental animal studies on the effects of micronutrients and macronutrients on offspring and human studies of pregnant women from cohorts with different nutritional conditions have contributed to current knowledge in the field of maternal nutrition. Diet and food choices define maternal nutritional status and directly influence fetal neurodevelopment [9]. Malnutrition, including undernutrition and overnutrition, can lead to altered maternal nutrient use. Fetal neurodevelopment is characterized by significant periods of rapid growth and structural change, accompanied by high metabolic and nutritional requirements. Thus, a fetus may be more susceptible to nutrient deficiencies and exposure to toxins depending on the timing, severity, frequency, and duration of exposure, as well as factors related to individual resilience [9].

The aim of this study was to evaluate how maternal nutrition, and specifically the intake of micronutrients, macronutrients, and vitamins in pregnant women, influence neurological development in offspring.

## 2. Materials and Methods

We performed a scoping review of the literature to assess how maternal nutrition can influence development of the nervous system in offspring. First, we conducted a systematic search of the literature to identify narrative or systematic reviews. The search was performed using the Pubmed database (https://pubmed.ncbi.nlm.nih.gov; accessed on 21 December 2020) with the following MeSH terms: ((“maternal nutrition”[All Fields] OR “maternal diet”[All Fields] OR (“maternal diet consumption”[All Fields] OR “maternal diet containing”[All Fields])) AND “fetal development”[All Fields]) OR (“fetal disease”[All Fields] OR “fetal diseases cerebrospinal”[All Fields]) OR “neurodevelopmental disorders”[MeSH Terms] OR “neurodevelopmental”[All Fields]. When searching for articles using ScienceDirect (https://sciencedirect.com; accessed on 21 December 2020), the terms used were adapted to (“maternal diet” OR “maternal nutrition”) AND (“fetal development” OR “fetal disease” OR “fetal disease cerebrospinal”). Additionally, we conducted a manual search to obtain articles listed in the reference lists of articles found in the initial search. Our search was limited to reviews published from 2000 until 2020. However, we included some classic articles published before 2000 because of their relevance. Additionally, we reviewed the references from each included paper to identify relevant citations, which we then retrieved manually. Exclusion criteria were articles published prior to 2000; articles written in languages other than English, Spanish, and French; articles related to purely animal evidence; as well as if they had a different design from narrative or systematic reviews.

The initial search retrieved a total of 3590 articles. After removing duplicates, the remaining articles were filtered by title and abstract (Figure 1). Then, the resulting 167 articles were randomly divided into five groups and assigned to an investigator, who analyzed the full texts to select articles for inclusion in the study. The results were discussed with all members of the team during weekly meetings. The full-text reviews led to the exclusion of 83 articles, leaving 84 studies for analysis.

We extracted the following data from the included articles: study design, study objective, characteristics of the nutrients and diets studied, dietary source of the nutrients, and main findings regarding the development of the nervous systems of offspring.

Considering the heterogeneity of the article types, we decided to present the results as a narrative review. We classified the articles into groups according to nutrient category as follows: type of diet, macronutrients, micronutrients, and others (Table 1).

## 3. Results

Most of the studies evaluated real-life needs regarding maternal supplementation of specific nutrients during pregnancy, and they considered the effects of imbalances in maternal nutrient levels on offspring neurodevelopment. These studies evaluated the importance of vitamins and minerals such as B12, folate, vitamin D, vitamin A, vitamin E, vitamin K, copper, iron, creatine, choline, zinc, and iodine; macronutrients such as polyunsaturated fatty acids (PUFAs) and proteins; generalized maternal nutrition status such as obesity, overnutrition (high fat (HFD) and hypercaloric diets), and undernutrition; and other nutrients with a relevant impact on offspring neurodevelopment such as gangliosides and caffeine. The main neurodevelopmental outcomes described were behavior and psychiatric disorders (autism spectrum disorder (ASD), attention-deficit/hyperactivity disorder (ADHD), schizophrenia, anxiety, depression, cognitive dysfunction (disabilities affecting intelligence, language, learning, and memory), visual impairment, motor deficits, neural tube defects, neural molecular dysfunction (such as that affecting synapses, myelin formation, neurotransmitter metabolism, cellular differentiation, plasticity, astrocytes, axonal and dendritic growth, anti-inflammatory regulation, vascular function, neuronal death, and oxidative stress), and finally, structural changes (such as diminished cerebral volume, spina bifida, hydrocephalia, and abnormal signaling in hypothalamic and hippocampal pathways), among others. Figure 2 shows the nutrients and neurodevelopmental outcomes that were most frequently encountered during data extraction.

## 4. Discussion

### 4.1. Diet

The maternal environment, including maternal diet and malnutrition during gestation, can have important consequences for offspring. Maternal malnutrition is a broad spectrum that includes HFD and hypercaloric diets, as well as caloric and nutrient restriction. All of these are considered early life stressors with the potential to predispose offspring to mental and physical disorders in adulthood [46]. Prenatal non-optimal nutrition may alter developmental processes such as synaptic plasticity, neurogenesis, and dendritic arborization [26].

Deleteric mechanisms have been proposed to arise from epigenetic changes that affect methylation in response to stress [46,75]. Concomitantly, a predisposition to conditions described by the neural modeling theory, as well as brain dysfunction due to early life exposure to malnutrition, can increase susceptibility to neuropsychiatric conditions such as ASD, depression, and schizophrenia [94]. For example, historical studies of the in utero effects of famine have indicated that an increased risk of schizophrenia is secondary to decreased intracranial volume and that low dietary intake of methyl groups such as choline, methionine, and folate alter the epigenomic profile of offspring [46]. One review suggested that maternal malnutrition led to permanently altered neuronal excitability, brain development, and cognitive and behavioral deficits in offspring as the result of a decrease in the number of hippocampal CA2, CA4, and DG neurons [46,75].

#### 4.1.1. Undernutrition

Brain development consumes over half of the available energy during gestation, and the main source of this energy is glucose from carbohydrates (75% of fetal energy) [10,47]. Thus, normal brain development is highly susceptible to nutrient restrictions, even when the fetal weight is normal [48]. Insufficient nutrient intake during the first stages of pregnancy alters neural cell proliferation, while undernutrition in the later stages affects neural differentiation [26,95]. Furthermore, maternal protein restriction has been associated with alterations in fetal astrocytogenesis, extracellular matrix structure, neuronal differentiation, mitochondrial function, death cell programming, expression of proteins related to the brain renin–angiotensin system (associated with hypertension), enhanced activity in the fetal hypothalamic–pituitary–adrenal axis (HPA), and hypomethylation in the ACE-1 brain gene promoter in animal models [46]. Furthermore, epigenetic changes in the brain are strongly related to maternal undernutrition. Specifically, maternal undernutrition was correlated with increased expression of the glucocorticoid receptor and proopiomelanocortin genes in the fetal hypothalamus, which is an important regulator of the HPA [46,50]. This could have various effects; for instance, the offspring of mothers who were undernourished were predisposed to altered food intake and imbalanced glucose homeostasis [51,52].

One study evaluated the effects of a 30% reduction in maternal nutrition on fetal baboons. The authors reported suppression of neurotrophic factor, imbalanced cell proliferation, and impaired glial maturation and neural synthesis, which were independent of maternal weight reduction [53]. Additionally, examinations of the effects of the 1944 Dutch famine on neurodevelopment in offspring born during that period found an increased rate of congenital central nervous system abnormalities, such as spina bifida and hydrocephalus. For instance, one magnetic resonance imaging study examining a 51-year-old schizophrenic man who was exposed to the famine during the first trimester of gestation reported predominantly white matter abnormalities [53,54]. Indeed, extreme nutrient deficiencies during development may increase the risk of schizophrenia, antisocial personality disorder, and admittance to addiction programs [25]. Other studies revealed the presence of hypo- and hyper-methylated gene loci in groups of prenatally undernourished individuals [55]. Neurocognitive consequences of cellular alterations caused by restricted fetal nutrition include impaired brain growth, poor visual recognition memory task performance, and delayed verbal ability. Poor maternal nutrition was also documented in children with intrauterine growth restriction (IUGR), and IUGR is associated with a lower IQ (a decrease of seven points at the age of seven) and schizophrenia in adulthood [11,26]. In summary, maternal undernutrition may affect the growth and structure of brain components such as the amygdalae, prefrontal cortex, hypothalamus, and autonomic nervous system and is related to impaired cognitive function and behavioral and psychiatric abnormalities [56,57].

#### 4.1.2. Overnutrition

##### High Fat Diet

Maternal HFDs have structural and functional neural effects on offspring that are related to diminished brain development. These effects include increased proliferation in the hypothalamus, as well as decreased apoptosis and neural differentiation in the dentate gyrus in animal models [26,27]. Furthermore, offspring exposed to a HFD during development are more susceptible to inflammatory alterations of the serotonergic neural system, which increases the risk of mental health and behavioral disorders such as anxiety, depression, ADHD, and ASD.

In animal models, mothers that received a HFD produced offspring with higher levels of inflammatory cytokines, which impacted neural development and manifested as anxious behavioral activity. However, Sasaki et al. reported that anxiety in these offspring was age-dependent, and the symptoms appeared to decrease with time [28,29]. A molecular hypothesis suggests that maternal HFD elevates levels of hypothalamic protein kinase C. In rats, the male offspring of mothers that received a 60% HFD were cognitively impaired and exhibited slower learning acquisition and memory retention [29]. In addition, a maternal HFD during gestation altered gene expression in offspring. Modifications were also observed in the dopaminergic systems of rodents after exposure to a maternal HFD, especially in the nucleus accumbens and ventral tegmental area, which are part of the reward circuit. These changes led to increased HFD consumption in offspring [30]. Molecular studies also showed that maternal hypercaloric diets led to the reprogramming of myeloid progenitor cells, which increased immune responses throughout the life span.

In human studies, inflammatory mechanisms have also been related to cognition in newborns exposed to HFDs during pregnancy. For example, a maternal diet that included lard was associated with altered spatial memory and learning abilities in adult offspring, and this was primarily mediated by microglial reactivity and decreased brain-derived neurotrophic factor (BDNF) in the hippocampus [30,31]. Furthermore, maternal HFD-related alterations in gene expression were related to the hypomethylation of opioid gene promoters in the brains of offspring, particularly in brain reward regions such as the ventral tegmental area, prefrontal cortex, and nucleus accumbens, and these changes increased the future obesity risk in the offspring [32,33]. Some studies have related these epigenetic changes and this pro-inflammatory profile with the promotion of microstructure and macrostructure changes that manifest as ASD symptoms and last through adulthood. Various clinical studies suggested that maternal obesity caused ASD in 1.39% and 1.59% of cases [34,35,36,37,38].

##### Obesity

Maternal obesity during pregnancy may induce immune-inflammatory changes that can affect the development of fetal neural pathways involved in behavioral regulation and cognitive performance. Among the described mechanisms of maternal hyperactivation of the HPA, glucocorticoid secretion and increased fetal cytokine expression were proposed to have an epigenetic effect on children, with an increase in psychological conditions and neurobehavioral morbidity. Some studies have suggested that high levels of IL-6 during pregnancy may be related to working memory dysfunction and dementia in offspring later in life. However, more research is needed to specify the cognitive consequences of maternal obesity [29,39,40,41]. Some studies have shown that these epigenetic changes have deleterious effects on serotonin and dopamine pathways, which are critical to behavior regulation [41,42]. Studies on the effects of maternal obesity on functional symptoms of attention deficit disorders, aggressive behavior, eating disorders, and psychotic disorders in adulthood have been inconclusive, indicating that more research is needed [40,41,42,43]. However, Sanchez et al. conducted a systematic review and reported that the offspring of mothers who were obese during pre-pregnancy were 17% more likely to have adverse neurodevelopmental outcomes compared with controls [41,44]. Specifically, the children of overweight mothers were 30% more likely to have ADHD, 10% more likely to have ASD, and 23% more likely to have intellectual developmental delays compared with children born to mothers with a normal weight [41,44]. The main proposed mechanisms for this effect were inflammatory and neuroendocrine changes [29]. However, De Capo et al. revealed that sucrose exposure induced hyperactivity, impulsivity, and altered attention and that these changes were secondary to altered striatal dopamine transport and receptor expression. Thus, their work elucidated one of the mechanisms for a relationship between maternal obesity and ADHD in offspring [13]. Other studies found no strong associations between body mass index (BMI) during pregnancy and ASD [29].

Specifically, high maternal consumption of fructose, present in soft drinks and processed foods, is directly related to metabolic syndrome and obesity due to increased production of triacylglycerols and branched chain amino acids [45]. Research suggests a relationship among high fructose intake and increased network hyperexcitability and altered prefrontal cortex function in offspring, as seen in ASD [12]. In animal models, high fructose consumption in corn syrup may be related to impaired hippocampus cognition due to hypermethylation and reduced expression of the BDNF gene [14]. Overall, it has been proposed that fructose may influence fetal brain development and predispose offspring to ASD [12].

Metabolic changes have been reported in children exposed to maternal obesity during development. For instance, high levels of leptin in obese mothers was associated with placental dysfunction and altered neural development in offspring [30]. Leptin, insulin, and ghrelin are able to cross the blood–brain barrier, and they act as regulators of neural plasticity via facilitation of glutamatergic and GABAergic activity, which promotes cognition [29,30]. Children with ASD have been found to have higher plasma levels of these hormones [30]. Maternal obesity is closely associated with gestational diabetes, hyperglycemia, and hyperinsulinemia, which may cause alterations in neural circuitry during periods of brain development [30]. In mice, the offspring of mothers that were obese during pregnancy exhibited insulin resistance as a marker of decreased neurogenesis and synaptic plasticity [29]. Among other mechanisms, eating a HFD during pregnancy has been associated with changes in HPA-axis activity and stress responses in offspring, which are related to abnormal brain signaling and behavior. This occurs via an increase in glucocorticoid receptor expression in the hippocampus, which plays a role in HPA inhibition [29].

Maternal obesity may be related to structural alterations, such as neural tube defects, as well as other congenital anomalies, like anencephaly and spina bifida. Compared with those with a normal BMI, obese mothers were more likely to produce offspring with structural defects. However, observational studies are insufficient to establish a strong causal relationship between maternal BMI and brain structural abnormalities [40]. 

##### Ketogenic Diet

The ketogenic diet is characterized by low consumption of carbohydrates and high intake of fatty acids. Barry et al. discussed the function of ketone bodies in neurodevelopment and suggested that they play a role in neurite formation [15]. In animal models, long-term maternal ketonemia reduced glucose uptake in the brains of offspring, generating larger brains and varied sizes of specific brain regions, such as the hippocampus, hypothalamus, and striatum. The long-term effects of increased ketone bodies during pregnancy are associated with hyperactivity in offspring, as well as symptoms of anxiety in adult offspring. Further studies are needed to establish a relationship between ketonemia during pregnancy and cognitive effects in offspring [15]. 

### 4.2. Macronutrients

#### 4.2.1. Fatty Acids

Dietary polyunsaturated fatty acids (PUFAs) are mainly found in oily fish, nuts, seeds, and green vegetables. PUFAs support the development of the fetal brain, nervous system, and retina and are particularly important during the third trimester of pregnancy [10,15]. They are involved in the formation of neuronal membranes, cell energy, synapse maturation, and myelination from the 24th week of gestation until childhood [5,13]. Docosahexaenoic acid (DHA) constitutes 90% of omega-3 PUFAs in the brain, and it is obtained via conversion of alpha-linoleic acid and linoleic acid. Fatty acids ω-3 and ω-6 also play a role in the regulation of levels of DHA, which is directly involved in brain function [10]. Maternal intake of DHA and PUFAs is fundamental to placental function, as well as dendritic growth and neural synaptogenesis, because the fetus is dependent on the mother’s supply until 16 weeks after birth [16,17].

Evidence from animal studies has shown that the inadequate intake of ω-3 fatty acids decreases DHA levels in the brain, leading to impaired neurogenesis, neurotransmitter metabolism (dopamine and serotonin), learning, and visual function and increased stereotyped behavior in primates. Some studies have suggested that changes in the fetal brain occur after a mother consumes a diet that is composed of more than 10% fat, with low levels of 18:3ω-3 and high levels of 18:2ω-6 fatty acids. Among other pathogenic mechanisms, ω-3 fatty acid deficiency is related to slower membrane fusion and Soluble NSF Attachment Protein Receptor protein dissociation in the hippocampus due to collateral DHA deficiency. In addition, a lack of maternal ω-3 fatty acid can lead to hippocampal underdevelopment due to impaired postmitotic cell migration, which is related to decreased embryonic cell growth. Some studies have even suggested that hydrogenated fat exposure during pregnancy in rats may decrease hippocampal ω-3 PUFA. Deficiency of ω-3 fatty acid may also decrease the size of neuronal cell bodies in other brain structures such as the parietal cortex and hypothalamus. These changes may be permanent, as they may affect neurogenesis, dendritic arborization, myelination, or synaptogenesis [18]. In contrast, in mice and rats, excessive maternal PUFAs may also be deleterious with respect to offspring development and has been associated with decreased hippocampal neurogenesis and altered signaling. Excessive PUFAs exposure during pregnancy may manifest as an altered stress response and avoidance of open spaces, known as thigmotactic behavior (considered an index of anxiety), in offspring [13,19]. Furthermore, 20:4ω-6 plays an indirect role in the regulation of brain development, especially in terms of synaptic plasticity, long-term potentiation, and spatial learning through prostaglandins synthesis. However, 22:5ω-6 cannot be substituted for DHA in neurite outgrowth and formation [18]. 

Evidence from human studies has shown that consumption of long-chain PUFAs during pregnancy is associated with greater benefits in specialized task performance in older children versus infants and younger children [11]. Low maternal DHA levels during pregnancy are associated with an increased risk of altered neural development. For example, some studies in the review by Innis showed a decreased risk of low visual acuity when maternal DHA intake was sufficient during pregnancy [18]. However, in that study, maternal 22:4ω-6 was considered to be a better predictor of visual acuity in infants because it contributes to neural development, even in the absence of DHA deficiency [18]. Despite this, a review by Jensen indicated that children whose mothers ate oily fish during pregnancy had higher levels of stereo acuity at 3.5 years of age, because of the relationship between HDA levels and the maturation of pattern-reversal visual-evoked potential and retinal sensibility [20].

Other studies have reported that normal levels of maternal ω-3 decreased the risk of lower IQ in offspring [20]. Furthermore, sufficient maternal DHA levels were associated with better mental and sequential processing scores at 4- and 7-years-old and higher verbal abilities and visual acuity [11,16]. In other experimental studies, children whose mothers ate diets with higher levels of cod liver oil or corn oil with DHA and eicosapentaenoic acid during gestation obtained higher scores on the Kaufman Assessment Battery for Children, compared with those whose mothers did not take these supplements [20]. Despite the findings of these previous studies, a meta-analysis by Gould et al. concluded that there were no significant differences in cognitive, language, or motor development between supplemented and non-supplemented groups [21].

In contrast, neurodevelopmental disorders have been related to higher dietary intake of PUFAs ω-6, low consumption of ω-3, and HDA. Higher levels of HDA appear to reduce the risk of schizophrenia, bipolar disorder, depression, and anxiety, among other diseases. One of the main risk factors for these conditions is the consumption of Western diets. These diets are characterized by meat and processed food, which often contain pro-inflammatory ω-6, and a lower intake of seafood, which contains the anti-inflammatory ω-3 [16]. Overall, the association between PUFAs and fish intake during pregnancy and the risk of ASD and ADHD in offspring is still inconclusive, indicating that more studies are needed. Contradictory findings in existing studies may be attributable to differences in the assessments of exposure in terms of maternal diet [5]. For example, in observational studies, seafood consumption and neuropsychological development in offspring during early childhood cannot be causally related because of additional factors that may influence child development [22]. 

One mechanism by which fatty acids may influence fetal development is the modification of DNA via the methylation of BDNF, nerve growth factor, and vascular endothelial growth factor, which is induced by ω-3 fatty acids [23]. BDNF plays a very important role in brain development, neurovascular function, and cognitive functions such as memory and learning. However, the proposed relationship between altered neurotrophin concentration and the risk of neurodevelopmental and behavioral disorders in offspring requires further examination [11,23].

Adequate PUFAs intake should be guaranteed through maternal diet or nutritional supplementation. Currently, pregnant women are encouraged to take a minimum of 200 mg/day of DHA. Martins et al. emphasized the importance of supplementation during pregnancy in mothers with dietary deficiencies [16,22]. There appears to be a positive relationship between maternal supplementation with DHA and infant status at birth in terms of latencies of visual-evoked responses, with an especially beneficial impact on neuromotor development [17]. Supplementation in combination with specific socioeconomic characteristics (such as maternal education), interaction with external factors, and a healthy lifestyle (such as non-smoking status), may improve DHA status during pregnancy [22]. For example, fat intake is higher in Latin American countries than in Africa and Asia, and fat intake in Western nations is often insufficient to support higher demands during pregnancy [22,24]. Overall, pregnant women are encouraged to engage in seafood consumption to prevent neurodevelopmental disadvantages in offspring [22].

In summary, the availability of fatty acids in the nervous system is very important for neurodevelopment. Infants should have adequate PUFA intake during development, and pregnant women should guarantee optimal levels of fatty acids through normal nutrition or supplementation. Among different forms of fatty acids, higher levels of DHA, arachidonic acid, and PUFAs are associated with better postnatal neurological outcomes, especially in terms of neuromotor abilities [17]. Nevertheless, more studies are needed to clarify the beneficial potential of maternal supplementation in terms of offspring neurodevelopment [22].

#### 4.2.2. Proteins

Proteins are found in meat, fish, eggs, legumes, nuts, and seeds, among other foods [10]. They are essential throughout gestation, especially during the second and third trimesters, due to accelerated fetal tissue growth. They are also important as an alternative source of energy when carbohydrate intake is deficient [10].

Some experimental studies have supported the idea that the children of mothers exposed to high-protein diets and energy beverages had better intellectual abilities in terms of information processing, numeracy, and vocabulary compared with control children [11]. Prado and Dewey conducted a review of maternal food supplementation and the associated effects on brain development in offspring [25]. They found positive effects of protein intake in terms of various motor and cognitive abilities. A daily intake of 45 g and 50 g of protein is recommended during pre-pregnancy and pregnancy, respectively [10]. Although studies have concluded that adequate nutrition is essential for cognitive and motor development, the optimal timing of nutrition supplementation is still undetermined [25].

In animal models, protein restriction during pregnancy increases the risk of heat aversion, stress sensitivity, and thigmotaxis in offspring. However, if protein restriction takes place prior to conception, biological maternal adaptation to the diet may occur, thus reducing the severity of behavioral alterations in offspring. In addition, studies found that the response to restrictive diets varied according to sex in animals. Specifically, males showed decreased avoidance behavior, and females expressed increased attitude abnormalities, expressed as depression-like symptoms [13]. Among the involved mechanisms, protein restriction during fetal development was found to modify hippocampal neurogenesis, which reduced BDNF and insulin-like growth factor and also decreased brain and neuronal volume. Finally, a lack of protein consumption was found to decrease maternal lipid availability and fetal brain fatty acid levels. This may reduce myelin production, with potential implications in terms of the risk of neuropsychiatric disorders [13].

### 4.3. Micronutrients

#### 4.3.1. Iron

Iron requirements increase during pregnancy, with average levels rising to 1000 mg to support metabolic and oxygenation processes [68]. Prophylactic iron supplementation during early pregnancy (i.e., 30 to 40 mg/day taken from the 20th week of gestation until delivery) can increase hemoglobin concentrations and body iron stores in pregnant women [58,68]. Fetal iron concentrations depend on maternal iron status in the absence of other pathologies [58,68]. Important developmental processes, such as myelination, dendritogenesis, synaptogenesis, and neurotransmission, are dependent on iron-containing enzymes and hemoproteins [69]. These processes could be affected by an iron deficiency depending on the gestational period, with different potential neurodevelopmental alterations [69]. For instance, iron deficiency during the first trimester can lead to alterations in gray matter and dendritic structure, triggering long-term effects that can manifest as memory alterations and neurodevelopmental impairment, including motor deficiencies, social dysfunction, and low academic performance [58,68]. Chronic fetal hypoxia, which is seen in IUGR and gestational diabetes, results in increased iron use to compensate for fetal erythropoiesis, and it may contribute to long-term developmental abnormalities [69].

Iglesias et al. assessed evidence regarding the role of prenatal iron on neurodevelopment and behavior, and they concluded that both excess and deficient iron can affect offspring [96]. In excess, iron may be toxic because of its ability to generate reactive oxygen species and to induce cell and tissue damage [58]. In contrast, poor fetal iron status is associated with reduced language ability, fine motor skills, and tractability, as well as decrements in measures of global intelligence, poor behavioral abilities, poor recognition memory, and slow brainstem-evoked response latencies [26,69,70]. Outcomes related to fetal iron deficiency persist despite postnatal iron repletion [58]. Although the connection between prenatal iron supplementation and ASD risk in offspring has been studied, evidence is limited. Furthermore, the available data do not suggest a clear association between these factors [5].

#### 4.3.2. Copper

Copper can be found in offal, nuts, cereals, fruits, and, to a lesser extent, milk and dairy products [80]. Although nutritionally-induced copper deficiency is rare in humans, pregnant women may have a low copper intake [80]. Copper supplementation during pregnancy has not been thoroughly examined [71]. Secondary copper deficiencies resulting from interactions with other nutrients or drugs, as well as congenital enzyme alterations, have been described. Studies with animal models have shown that copper deficits during pregnancy are related to embryonic death, as well as structural, biochemical, neurological, and immunological abnormalities [82].

#### 4.3.3. Creatine

The daily creatine requirement in adults is around 2 g; creatine can be obtained as part of an omnivorous diet through animal-derived products or supplements. Infants obtain this micronutrient from breast-feeding or milk-based formulas. Thus, vegans and infants fed with non-dairy formulas may not receive the required amount [83]. Creatine is also endogenously synthesized in the liver. It has an important role in maintaining cellular ATP levels and stabilizing the mitochondrial membrane potential. Animal studies have demonstrated that in a mouse model of fetal asphyxia, creatine supplementation mitigated hypoxia-related motor alterations in male spiny mice [76]. However, few studies have assessed neurological outcomes following creatine supplementation in pregnant women.

#### 4.3.4. Choline

Exogenously, choline can be obtained mainly from animal-based foods and, to a lesser extent, from plant-based foods. It can also be produced endogenously in the maternal liver [77]. Choline deficiency is common, and it is estimated that 90% of pregnant women in the USA do not meet recommended intake levels [77]. This micronutrient is implicated in various molecular pathways, including phospholipid and neurotransmitter synthesis. It acts as a methyl donor, inducing epigenetic modifications in the fetal brain and placenta, and it is involved in stem cell proliferation and transmembrane signaling during neurogenesis [25,76].

Choline has an important role in neurodevelopment and is implicated in the adaptive modulation of cognitive functions [46]. Maternal choline deficiency alters neurogenesis and angiogenesis in the fetal hippocampus [46,75]. Normal choline concentrations increase the number and size of cholinergic neurons in the medial septum [77]. Human studies evaluating maternal choline supplementation have produced mixed results. Ross et al. indicated that oral choline supplementation during the second trimester and after birth was associated with better sensory gating [78]. Some studies have suggested that maternal choline intake and maternal choline concentrations are inversely associated with the risk of neural tube defects in offspring [25,77]. Other studies failed to demonstrate an effect of maternal choline supplementation on intelligence or infant cognition [76].

#### 4.3.5. Zinc

Mild to moderate zinc deficiency is estimated to affect 30% of the global population. Plant-based diets contribute to zinc deficiency by two mechanisms [79]. The first is low intake, because meat and especially shellfish are the richest sources of this micronutrient, and the second is through the inhibition of zinc absorption by fiber and phytates. Zinc plays an important role in fetal development because of its role in carbohydrate and protein metabolism, nucleic acid synthesis, cell division, and differentiation [79]. Rodent studies have indicated that gestational zinc deficiency is associated with decreased cell counts and reduced regional brain mass in the cerebellum, limbic system, and cerebral cortex [11]. Although the fetuses of zinc-deficient mothers have shown decreased movement, enhanced heart-rate variability, and alterations in autonomic nervous system stability, the available evidence does not strongly support the idea that supplementation improves childhood cognitive or motor development [11].

#### 4.3.6. Iodine

Because of its role in thyroid hormone synthesis, iodine is a relevant nutrient in fetal neurodevelopment. The first half of gestation is dependent on the thyroid hormone, and maternal T4 is essential for neuronal migration and myelination of the fetal brain. Indeed, irreversible neurological damage develops in the absence of the thyroid hormone [72]. Furthermore, maternal iodine deficiency during pregnancy can cause iodine deficiency-related disorders in offspring [71]. Deficient iodine has been found to disrupt fetal neurogenesis, neuronal migration, synaptogenesis, and myelination [9]. Neurological outcomes described include congenital anomalies, endemic cretinism, and subclinical cognitive and motor function deficits [71]. Indeed, iodine deficiency has been described as the leading cause of preventable impaired mental function worldwide. There is a wide spectrum of clinical manifestations, depending on the deficiency severity, such that pregnant women with a mild degree of iodine deficiency are less likely to have children with more severe neurocognitive impairment, including learning difficulties and decreased IQ scores [73]. In contrast, severe iodine deficiency in pregnancy manifests as strongly impaired cognition in offspring. Cretinism is described as the most severe clinical manifestation, and it is characterized by hearing, speech, and gait alterations, as well as low IQ scores [74].

Current evidence supports iodine supplementation during pregnancy, with higher rates of effectiveness in preventing neurological damage if started before conception or in the first trimester. Continuation is recommended throughout pregnancy, considering the span of action of thyroid hormone in the fetal brain [73]. Multiple medical societies and governments recommend iodine supplementation during pregnancy. Although the actual suggested intake is 250 mg/day, the true intake depends on household consumption of iodized salt [72]. Other dietary sources of iodine include sea fish, shellfish, cereals, and grains [10].

#### 4.3.7. Vitamin B12

Vitamin B12 acts as an enzyme and cofactor. It mediates mitochondrial succinil-CoA formation and cytosol methionine synthesis and is essential for fat and protein metabolism, as well as hemoglobin generation. Furthermore, vitamin B12 contributes to DNA methylation and epinephrine synthesis [9,63]. Sufficient vitamin B12 levels are required for normal neuronal development and myelination [11].

As B12 is mainly found in animal products, vegan and vegetarian diets enhance the risk of B12 deficiency in pregnant women. Indeed, case reports of women with vegan diets during pregnancy and B12-related deficiency have reported that the offspring fail to thrive, exhibit irritability, and show reduced cerebral growth [63]. Maternal B12 deficiency, i.e., values below 200 pg/mL, has also been associated with an increased risk of neural tube defects [58].

#### 4.3.8. Folate

Folic acid is a precursor in the metabolism of amino acids and nucleic acids. It is estimated to be involved in more than 100 metabolic reactions and multiple regulation pathways [58,59]. Inadequate folate intake has been associated with altered fetal brain development through incorrect DNA methylation [5]. Folate adequate levels are required for neural cell proliferation, migration, differentiation, vesicular transport, and synaptic plasticity [60,61]. The role of folate supplementation in protecting against neural tube defects has been widely described. This protective effect appears to occur because folate promotes neurogenesis and has axonal pro-regenerative effects, as observed in rodent models [60]. Some studies have supported a role of folate in cognitive function and learning ability; however, the mechanisms by which this might happen remain unclear. Other studies have shown that folic acid supplementation leads to enhanced vocabulary development, communication skills, and verbal comprehension at 18 months [60,61]. Furthermore, an association between maternal folate intake and an enhanced risk of ASD has been described, with the identification of the MTHFR 6777 C > T variant [60,62]. Folate requirements during pregnancy increase by around 50%. For these requirements to be met, periconceptional supplementation of 400 mg/day is recommended [58].

#### 4.3.9. Vitamin D

Vitamin D affects brain development through its ability to regulate gene expression [65]. Prenatal deficiency results in abnormal brain development, persistent changes in adult brain structure, neuronal differentiation, neurotransmission, synaptic plasticity, axonal connectivity, neurochemistry, and dopamine ontogeny, as well as changes in other biological pathways, including those implicated in oxidative phosphorylation, cytoskeleton maintenance, calcium homeostasis, chaperoning, and post-translational modifications, which indicates that vitamin D plays a role in epigenetic regulation [46,62,66]. Current evidence supports the relationship between prenatal vitamin D levels and an enhanced risk of ADHD and ASD, indicating that it impacts neuropsychological development in children [66]. Villalobos et al. performed a meta-analysis, and they found that children born to vitamin D-insufficient mothers showed poorer mental and language development [64]. Current recommendations for pregnant women include a daily intake of 600 UL of vitamin D, with dietary sources of vitamin D including fatty fish (such as catfish, salmon, mackerel, and tuna), as well as egg, beef, and fish liver oil [66].

#### 4.3.10. Vitamin A

Vitamin A (retinoic acid) and its metabolites are implicated in neural patterning, neuronal differentiation, neurite outgrowth, and axonal elongation [44]. Retinoic acid is mainly synthesized in the embryo, regulates expression of developmental target genes, and directs organogenesis [67]. The intake recommendation for pregnant women is 600 mg, with eggs, oily fish, fortified low-fat spreads, milk, and yogurt as the main dietary sources. High levels of vitamin A should be avoided considering its teratogenic effects [10].

#### 4.3.11. Vitamins E and K

There is limited evidence regarding the role of maternal vitamin E and K supplementation in fetal neurodevelopment.

### 4.4. Other Elements

#### 4.4.1. Gangliosides

Gangliosides constitute 6% of the phospholipids in the nervous system and may play a crucial role in brain development. There are several ganglioside subtypes (GM1, GD1a, GD1b, GT1a, and GT1b), and the concentrations of these subtypes in the brain change during neural proliferation and maturation [92,97,98]. Gangliosides are regulated by lysosome and endosome metabolism and can determine neural repair, some neurological diseases, ion channel modulation, neurotransmitter release, and signal transduction, among other functions in the body [88,92,93]. Egg yolk, meat, and milk are the only sources of exogenous gangliosides. Dietary supplementation during pregnancy may have long-term effects on brain development, and some studies have supported the idea that increasing the total maternal dietary intake of gangliosides by 1% could enhance cognitive development in offspring [92].

During fetal development, gangliosides are predominantly concentrated in the hippocampal region, in which ganglioside action is related to cognitive function and memory. GM1, GD1a, and GT1b are very important for the maintenance of myelin in the nervous system. Changes in the concentrations of these compounds are directly related to glial and neural proliferation. For example, between weeks 8 and 25 of gestation, the expression of CD3 and Gm3 is high in neuronal and glial precursor cells [89,90,91]. Accordingly, maternal dietary supplementation with gangliosides may have a positive effect on fetal brain development [92].

#### 4.4.2. Caffeine

Caffeine may impact fetal neurodevelopment via physiological changes during pregnancy. When caffeine crosses the placenta, the metabolic rate reduces such that it has a subsequent half-life of 2 h or longer (4.5 to 15 h). Since the fetus does not have enzymes for its metabolism, caffeine may affect gestational health [84]. Some animal studies have reported that maternal caffeine consumption is associated with altered sleep, locomotion, learning abilities, and anxiety and that it may interfere with brain zinc fixation, especially in mothers with a low protein diet [85]. However, human studies have reported that an increased risk of hyperkinetic disorders and ADHD only occurred in children exposed in early pregnancy to more than 10 cups of coffee per day and that less maternal intake was not related to any risk [5,86]. Qian et al. examined the EDEN 1083 Mother–Child Cohort, in which the offspring of mothers with a caffeine intake higher than 200 mg per day had a twofold higher risk of altered cognitive development and IQ impairment at 5.5-years-old, compared to children who were exposed to less than 100 mg of caffeine per day [87]. More research is needed to confirm a strong association between neurodevelopmental conditions in offspring and extreme levels of maternal caffeine consumption (higher than 1000 mg/day or 10 cups/day) [5,86].

## 5. Conclusions

An unsupplemented maternal diet may be insufficient in terms of the required nutrients for optimal fetal health. Furthermore, mothers who eat vegan diets tend to have a reduced intake of several important compounds (such as zinc and creatine). Researchers have established inadequate micronutrient intake levels (deficiency and excess) and their outcomes in terms of neurodevelopment. For instance, iron has been implicated in neurogenesis, and iron deficiencies have been associated with memory alterations and neurodevelopment impairment. These outcomes persist even after postnatal iron repletion. Furthermore, iodine deficiency has been described as the leading cause of preventable impaired mental function worldwide, with outcomes ranging from congenital anomalies and endemic cretinism to subclinical cognitive and motor function deficits. Folate supplementation is known to play a role in protecting against neural tube defects, and it is implicated in cognitive function and learning abilities. Vitamin A is crucial for organogenesis, neural patterning, neuronal differentiation, neurite outgrowth, and axonal elongation. It is of special consideration that levels above 10,000 IU have been related to teratogenesis. Choline deficiency is common during pregnancy, and it is known to alter neurogenesis and angiogenesis in the fetal hippocampus. Furthermore, some studies have shown an association between choline deficiency and neural tube defects. Copper and creatine deficiencies during pregnancy are rare, and evidence regarding their role in neurodevelopment is scarce. Considering the effects of these compounds on physiological adaptation and fetal developmental processes, micronutrient supplementation is recommended during pregnancy.

Prenatal maternal intake of macronutrients plays a significant role in offspring neurodevelopment. Fatty acids, especially PUFAs (Omega-3 and Omega-6) and DHA, are strongly related to neuronal functioning, such as synapse maturation and myelination during development. Insufficient fatty acids during development can increase the risk of low IQ; altered visual acuity; and altered cognitive, language, and motor development. Furthermore, higher levels of DHA, arachidonic acid, and PUFAs are associated with better neurological postnatal outcomes, especially in terms of neuromotor abilities. However, supplementation is still under debate. Protein consumption and its effect on neurodevelopment is also unclear, although some studies have suggested a positive relationship between adequate protein consumption and motor development. Furthermore, decreased protein consumption might be related to neuropsychiatric disorders.

Finally, different types of diets during gestation may have consequences on offspring neurodevelopment. Undernutrition during gestation has been related to the altered growth and maturation of brain structures, especially the amygdalae, prefrontal cortex, hypothalamus, and autonomic nervous system. In contrast, overnutrition, which is related with high fat and carbohydrate diets, is related to inflammatory processes that can be deleterious to offspring brain development. These types of diets are associated with cognitive impairment and neuropsychiatric disorders, such as depression, ADHD, ASD, and anxiety. More studies are needed to address the benefits and risks of supplementation of each nutrient during pregnancy, with the goal of establishing strong recommendations. Overall, inadequate maternal intake of micronutrients and macronutrients can have significant short- and long-term effects on neurodevelopment in offspring.

## Figures and Tables

**Figure 1 nutrients-13-03530-f001:**
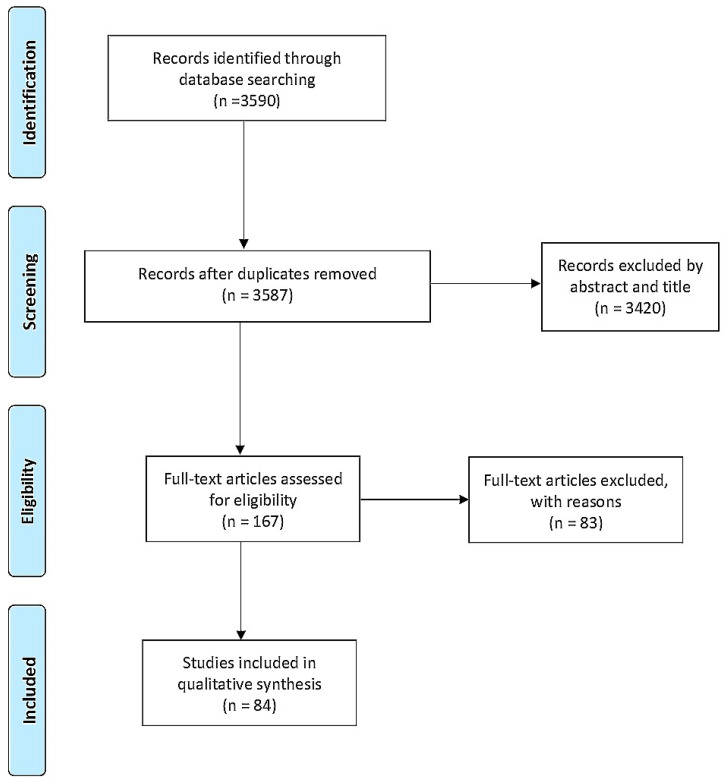
Prisma Flow Diagram.

**Figure 2 nutrients-13-03530-f002:**
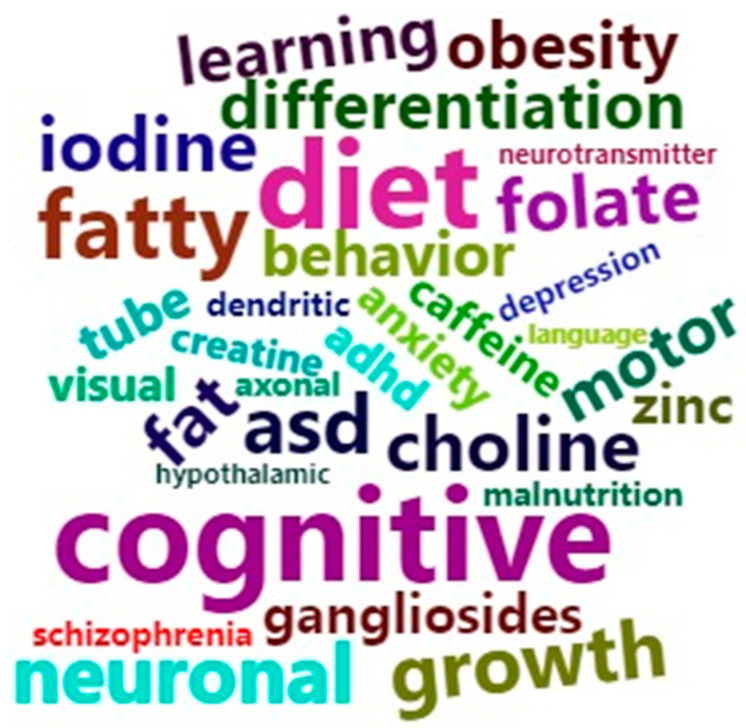
Graphical representation of the nutrients and neurodevelopmental outcomes most frequently encountered during the review. The chosen terms were based on the extraction of data from all included articles using the R cloud program.

**Table 1 nutrients-13-03530-t001:** Nutrients evaluated.

Nutrient Category		Nutrient	Reference Citation
**Macronutrients**		Fatty acids	[5,10,11,12,13,14,15,16,17,18,19,20,21,22,23,24]
		Proteins	[10,11,13,25]
**Diets**	**Overnutrition**	High fat diet	[26,27,28,29,30,31,32,33,34,35,36,37,38]
		Obesity	[12,13,14,29,30,39,40,41,42,43,44,45]
		Ketogenic diet	[15]
	**Undernutrition**	Maternal malnutrition/famine	[10,11,25,26,46,47,48,49,50,51,52,53,54,55,56,57]
	**Vitamins**	Folate	[5,58,59,60,61,62]
		B12	[9,11,58,63]
		Vitamin D	[46,62,64,65,66]
		Vitamin A	[10,44,67]
		Vitamin E	
		Vitamin K	
**Micronutrients**		Iron	[5,26,58,68,69,70]
		Iodine	[9,10,71,72,73,74]
		Choline	[25,46,75,76,77,78]
		Zinc	[11,79]
		Copper	[71,80,81,82]
		Creatine	[76,83]
**Other**		Caffeine	[5,84,85,86,87]
		Gangliosides	[88,89,90,91,92,93]

## Data Availability

Not applicable.

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
