# Peer review of "Maternal Nutrition and Neurodevelopment: A Scoping Review"

_nutrients, 2021, doi:10.3390/nu13103530_

Round 1
Reviewer 1 Report
This manuscript is a well-written review article to introduce the influence of maternal nutrition influence on offspring’s’ neurological development and disorders. The manuscript is informative and worthy to read. There are some suggestions for this manuscript.
- The criterial for article exclusion should be included in the section 2.
- More information should be included in table 1 with echoing the description presented in the Result section. Such as the linkage between mentioned nutrients and their reference citations.
- Food additives, such as fructose, have been reported to involved neurological hypertension and alteration of spatial learning in the adulthood. To discuss the relationship between maternal fructose intake and offspring’s neurological developmental programing is suggested to be included in this manuscript.
- It would be good to add a table to clearly show that have animal studies or clinical observations been reported to demonstrate the association between offspring’s neurological disorders and each maternal nutrient intake.
5. There is are some typo in the manuscript, such as that in line 176.
Author Response
Dear Reviewer,
We are very grateful for your recommendations to improve our article. Below, we will give answer to every one of them in the Word Document.

Reviewer 2 Report
This is a very well referenced and written manuscript describing the relationship between aspects of maternal nutrition and neurodevelopment in the offspring. Calcium and vitamin B6, two nutrients that may be inadequate in a mother's diet, were not included in the analysis. Why?
Author Response
Dear Reviewer,
We are very grateful for your recommendation to improve our article. In the word document, we will answer the question made.
